# PolarNet: 3D Point Clouds for Language-Guided Robotic Manipulation

**Shizhe Chen**,* **Ricardo Garcia***, **Cordelia Schmid, Ivan Laptev**
Inria, École normale supérieure, CNRS, PSL Research University
https://www.di.ens.fr/willow/research/polarnet/

**Abstract:** The ability for robots to comprehend and execute manipulation tasks based on natural language instructions is a long-term goal in robotics. The dominant approaches for language-guided manipulation use 2D image representations, which face difficulties in combining multi-view cameras and inferring precise 3D positions and relationships. To address these limitations, we propose a 3D point cloud based policy called PolarNet for language-guided manipulation. It leverages carefully designed point cloud inputs, efficient point cloud encoders, and multimodal transformers to learn 3D point cloud representations and integrate them with language instructions for action prediction. PolarNet is shown to be effective and data efficient in a variety of experiments conducted on the RLBench benchmark. It outperforms state-of-the-art 2D and 3D approaches in both single-task and multi-task learning. It also achieves promising results on a real robot.

**Keywords:** Robotic manipulation, 3D point clouds, language-guided policy

## 1 Introduction

People are able to perform a wide range of complex manipulation tasks in 3D physical environments. One efficient and intuitive way to specify different tasks is through natural language instructions. Consequently, a long-term goal in robotics is to develop robots that can follow language instructions to perform various manipulation tasks.

Most existing work on language-guided robotic manipulation uses 2D image representations [1, 2, 3, 4]. BC-Z [1] applies ResNet [5] to encode a single-view image for action prediction. Hiveformer [3] employs transformers [6] to jointly encode multi-view images and all the history. Recent advances in vision and language learning [7, 8] have further paved the way in image-based manipulation [4]. CLIPort [4] and InstructRL [9] take advantage of pretrained vision-and-language models [8, 10] to improve generalization in multi-task manipulation. GATO [11] and PALM-E [12] jointly train robotic tasks with massive web image-text data for better representation and task reasoning.

Although 2D image-based policies have achieved promising results, they have inherent limitations for manipulation in the 3D world. First, they do not take full advantage of multi-view cameras for visual occlusion reasoning, as multi-view images are not explicitly aligned with each other, as shown in Figure 1. Second, accurately inferring the precise 3D positions and spatial relations [13] from 2D images is a significant challenge. Current 2D approaches mainly rely on extensive pretraining and sufficient in-domain data to achieve satisfactory performance.

To overcome the limitations of 2D-based manipulation policy learning, recent research has turned to 3D-based approaches. The use of 3D representations offers a natural way to fuse multi-view observations [14] and facilitates more accurate 3D localization [15], see Figure 1 (right). For example, PerAct [14] adopts an action-centric approach that takes a high-dimensional input of over 1 million voxels to classify the next active voxel, achieving state-of-the-art results for multi-task language-

---

*The authors contributed equally to this work.

7th Conference on Robot Learning (CoRL 2023), Atlanta, USA.

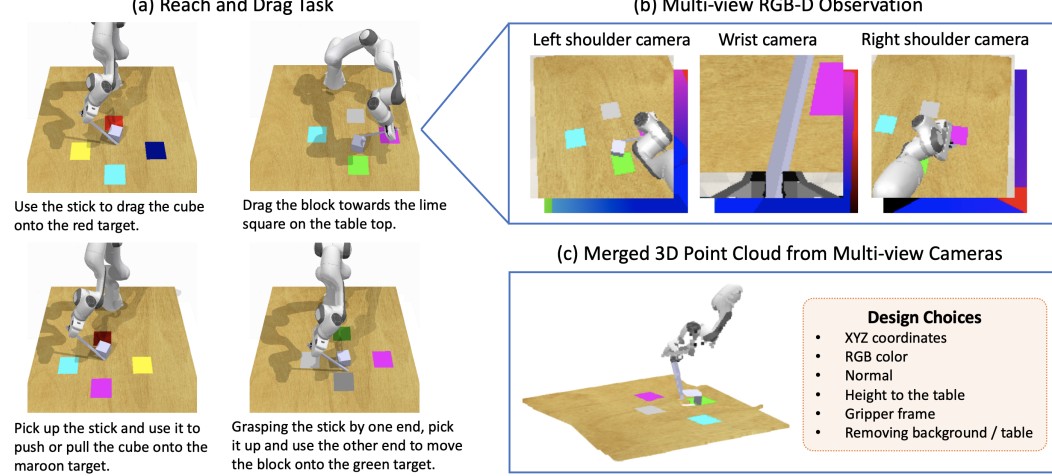

**Figure 1: (a)**: Variations of the "Reach and Drag" task in RLBench [19], with different target colors per variation. **(b)**: Although different views are complementary to represent the scene, they are not explicitly aligned with each other. **(c)** Merging multi-view cameras to construct a unified point cloud in 3D space. Design options of the point cloud input are carefully investigated in this work.

guided manipulation. However, such action-centric 3D voxels suffer from quantization errors and computational inefficiency. Alternative 3D representations in the form of point clouds have been successfully explored for 3D object detection [16], segmentation [17] and grounding [13]. However, the effective and efficient processing of 3D point clouds for robotic manipulation [18] is still underexplored. In addition, the focus has primarily been on single-task manipulation, lacking the versatility to incorporate language instructions for completing multiple tasks simultaneously.

This paper proposes **PolarNet**, a **po**int cloud-based **la**nguage-guided **r**obotic manipulation **net**work. We explore various design options for 3D point cloud representation, including the composition of point cloud features, the coverage space of the point cloud, the use of multiple camera views, and the choice of coordinate frames for point clouds. Our method builds on the PointNext architecture [17] to efficiently encode point cloud inputs. A multimodal transformer is employed to fuse the encoded point cloud with language instructions at an intermediate level. Finally, our PolarNet predicts 7-DoF (degrees of freedom) actions encompassing position, rotation and open state of a parallel gripper, which are executed by a motion planer to complete the task.

We conduct extensive experiments on RLBench [19] in three language-guided manipulation setups: single-task single-variation (one policy for each 10 tasks), multi-task single-variation (one policy for all 10 tasks), and multi-task multi-variation (one policy for 18 tasks with 249 variations). Figure 1 illustrates variations for the task "reach and drag". Our model significantly outperforms state-of-the-art 2D and 3D based models for all the three settings, demonstrating the effectiveness of PolarNet in integrating language guidance with a 3D point cloud based representation for robotic manipulation. PolarNet also achieves 60% average multi-task success rate on 7 tasks on the real robot.

## 2 Related Work

**Language-guided Robotic Manipulation.** Language-guided policy learning has gained significant attention in robotics due to its potential for convenient human-robot interaction and skill generalization across diverse tasks [20, 21, 22, 23, 24, 25, 26, 27]. Several multi-task benchmarks have been established recently [28, 19, 29, 30, 31]. In our work, we use RLBench [19] since it provides hundreds of challenging manipulation tasks with demonstrations and instructions. Leveraging pretrained large language models [12, 32, 33, 3, 4, 9, 11] has shown to be beneficial for language-guided manipulation. Most existing works are based on 2D images [2, 34, 1, 27, 3], while a few recent works explore the potential of 3D representations [14, 35]. Both C2F-ARM [35] and PerAct [14] utilize a

3D voxel representation. C2F-ARM proposes to represent the 3D scene in a coarse-to-fine manner to balance quantization error and computation cost. PerAct [14] uses an action-centric representation that encodes the language and all 3D voxels in the workspace via a Perceiver transformer [36] and classifies which voxel is the next gripper position, making it less efficient. In contrast, we explore the point cloud representations and predict actions in a more efficient "point-centric" manner.

**Manipulation Learning with Point Clouds.** Due to the advantages of 3D representations, research has increasingly focused on vision-guided robotic manipulation from 3D point clouds [37, 18, 38, 39, 40, 41, 42, 43, 44]. The design choice of the point cloud representation plays an important role for the task performance. Strudel et al. [45] shows the benefits from normals for motion planning. Liu et al. [18] finds that normalizing point clouds in the end-effector and target object frames work significantly better than the vanilla world and robot base frames. Seita et al. [46] adds per-point flow vectors to improve manipulation of tools. Qin et al. [47] further shows direct sim-to-real transfer is possible with point clouds. However, existing work mainly utilizes point clouds to train a single-task policy and does not consider any language interactions. In this work, we jointly represent point clouds and language to perform various manipulation tasks, and systematically evaluate design choices for point cloud representations, such as the importance of colors.

**Point Cloud Networks.** Point cloud-based vision approaches have studied the question of 3D representations for 3D object detection [16], segmentation [48] and 3D object grounding [13, 15]. PointNet [48] and its extension PointNet++ [49] are the most popular networks for processing point clouds. Although more advanced networks based on transformers [6] have been proposed recently [50, 51], PointNext [17] demonstrates that small modifications on PointNet++ can even outperform the newer architectures. Our approach PolarNet is based on the state-of-the-art PointNext model [17] and evaluates different design choices in the context of language-guided manipulation.

## 3 Method

### 3.1 Problem Definition

We aim to learn a visual policy $\pi(a_t|O_t)$ to perform robotic manipulation tasks following natural language instructions, where $O_t \in O, a_t \in A$ are the observation and action at step $t$, $O$ and $A$ are the observation and action space respectively. In this work, the observation space $O$ includes: 1) a language instruction $(x_1, \cdots, x_{N_x})$ where $x_i$ is a tokenized word, 2) RGB images from $K$ cameras $\{I_{rgb}^k\}_{k=1}^K, I_{rgb}^k \in \mathbb{R}^{H \times W \times 3}$, and 3) aligned depth images $\{I_{dep}^k\}_{k=1}^K, I_{dep}^k \in \mathbb{R}^{H \times W}$. We use $K = 3$ with cameras on the left shoulder, right shoulder and wrist of the robotic arm and $H = W = 128$ following previous work [2, 3]. The intrinsic and extrinsic camera parameters are known. The action space $A$ consists of the pose and open state of the parallel gripper. The pose is composed of the Cartesian coordinates $a_t^{xyz} \in \mathbb{R}^3$ and rotation described with a quaternion $a_t^q \in \mathbb{R}^4$ in the world frame. The open state $a_t^o \in \{0, 1\}$ indicates whether the gripper is open or closed. We follow the standard setup in RLBench [34, 2, 3, 14] to predict action of key steps and use a motion planner to execute the action. More details about the key step are presented in Section B of the appendix.

### 3.2 PolarNet: Point Cloud-based Language-guided Robotic Manipulation Policy

Figure 2 presents an overview of PolarNet for language-guided manipulation with 3D point clouds. More detailed architecture is presented in Figure 6 in the appendix. We first describe the design choice of point cloud representations in Sec 3.2.1, and then introduce the details of the model architecture in Sec 3.2.2 followed by the training objectives in Sec 3.2.3.

#### 3.2.1 Point Cloud Inputs

**Preprocessing.** Given $I_{dep}^k$ together with known camera instrincs and extrinsics at step $t$, we project each pixel in $I_{dep}^k$ to the 3D world coordinates. Since $I_{rgb}^k$ is aligned with $I_{dep}^k$, the RGB color of a pixel can be appended to the corresponding point. In this way, we obtain a point cloud $V_t^k \in \mathbb{R}^{H \times W \times 6}$

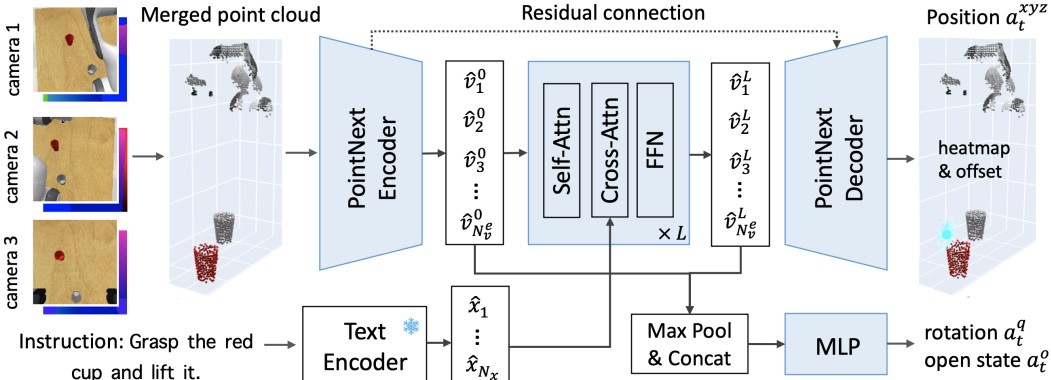

**Figure 2: PolarNet for language-guided manipulation.** The approach takes as input the merged point cloud obtained from multi-view RGB-D images and a language instruction, and uses Point-Next [17] for efficient point cloud encoding and CLIP text encoder [8] for language encoding. The point cloud and language are integrated via a multi-layer transformer at the intermediate level. PolarNet predicts the position (cyan node) using an integral over the heatmap of the point cloud with offset per point, and also rotation and open state of the gripper using global features.

for each camera in the world frame, where each point consists of the XYZ coordinates and its RGB color. It is possible to merge point clouds from different cameras, as they are in the same coordinate system. To reduce redundancy, we use Open3D toolkit [52] to uniformly downsample the merged point cloud to one point per voxel. We use a voxel size of $1cm^3$ following previous work [14]. We also estimate the normal of each point via Open3D. The geometric structure of the point cloud makes it straightforward to select relevant points, which is difficult for 2D images. For example, the background points like wall and floor are far away from the robot's workspace, and the manipulated objects are usually above the table surface[1]. Therefore, we define a 3D bounding box that covers the workspace above the table to crop the point cloud (Figure 5 in the appendix), only keeping points of the objects and the robotic arm. Our empirical results in Table 2 show the effectiveness of removing irrelevant points. We randomly sample $N_v = 2048$ points in the processed point cloud as $V_t$.

**Point cloud features.** The type of point cloud representation [45] and normalization [18] can significantly impact the performance. Our paper is the first to study this impact systematically. In the experimental section 4.2, we first investigate different compositions of the point cloud inputs, including: 1) XYZ coordinates, 2) RGB color which is seldom used for robotic manipulation because most works focus on a single task requiring geometry structures only to be solved, 3) normal which has been shown effective to avoid obstacles [45], and 4) the height with respect to the manipulation table which helps infer the object location. We then compare two types of coordinate frame to normalize the point cloud into a unit ball: 1) using the gripper position as the coordinate origin which has the benefits to infer the object-robot relation [18], and 2) using the center of the point cloud as the origin which is a common practice in 3D scene understanding tasks [15]. Our empirical results in Table 1 and 2 indicate that all the four point cloud representations are beneficial with color of particular importance, while the two coordinate frames perform similarly.

### 3.2.2 Model Architecture

**Language encoding.** We employ the language encoder from the CLIP model [8] to tokenize and encode the language instructions. It is pre-trained on large-scale image-text pairs, enabling to effectively understand visually relevant instructions. We freeze the language encoder and add a linear layer to it to obtain the language embeddings $\hat{X} = (\hat{x}_1, \cdots, \hat{x}_{N_x}) = W_x \, \text{CLIP}([x_1, \cdots, x_{N_x}])$.

**Point cloud encoding.** We encode the point cloud $V_t$ with the state-of-the-art PointNext encoder [17]. It consists of $L_e$ number of set abstraction (SA) blocks to hierarchically abstracts features

---

[1]There are few exceptions such as the "sliding block" task where the target is a green area on the table. We manually select those tasks and do not remove the table points for them.

of $V_t$. Each SA block contains a subsampling layer, a grouping layer, a multilayer perceptron (MLP), and a reduction layer. The subsampling layer at the $l$-th block uses farthest point sampling to select $N_v^l$ points from $V_t$, and then the grouping layer finds neighbors for each selected point which are points within a radius $r^l$ to the query. The MLP is shared for all the points to generate features. Finally, the reduction layer utilizes max pooling to aggregate features within the neighbors for the $N_v^l$ query points. Assume $v_i^l$ is the input feature of query point $i$ at the $l$-th block and $p_j^l$ is the XYZ coordinates of the point, the computation is formulated as:

$$v_i^{l+1} = \text{MaxPool}_{j:(i,j)\in\mathcal{N}}\{\text{MLP}([v_j^l; (p_j^l - p_i^l)/r^l])\}, \tag{1}$$

where $\mathcal{N}$ denotes the set of neighbor points to the query point $i$, and $[\cdot]$ is the feature concatenation. Each SA block decreases the number of points by 2 times, while increases the dimensionality of the features by 2 times. We denote the encoded point cloud as $V_t^{L_e} = \{v_i^{L_e}\}_{i=1}^{N_v^e}$ where $N_v^e = N_v/2^{L_e}, v_i^{L_e} \in \mathbb{R}^{2^{L_e}d_e}$ and $d_e$ is the hidden size in the first layer.

**Vision-and-language fusion.** We utilize a multi-layer transformer architecture [53] to integrate the point cloud features $V_t^{L_e}$ and the language embeddings $\hat{X}$. To be specific, we first add the XYZ position and step id embedding to the point cloud feature as follows:

$$\hat{v}_i^0 = W_v\, v_i^{L_e} + W_p\, p_i^{L_e} + E_s(t), \tag{2}$$

where $E_s(\cdot)$ is the sinusoidal positional encoding [6] to denote different steps. Then, in each transformer layer, we sequentially perform self-attention on the point cloud features, cross-attention from the point cloud to the language followed by a feedforward network, which is:

$$\text{Attn}(Q,K,V) = \text{Softmax}\left(\frac{W_Q Q (W_K K)^T}{\sqrt{d}}\right)W_V V, \tag{3}$$

$$\hat{V}'^l = \text{Attn}(\hat{V}^l, \hat{V}^l, \hat{V}^l), \quad \hat{V}''^l = \text{Attn}(\hat{V}'^l, \hat{X}, \hat{X}), \quad \hat{V}^{l+1} = \text{LN}(W_2\,\text{GeLU}(W_1\hat{V}''^l), \tag{4}$$

where $d$ is the hidden size in the transformer layers and LN denotes layer normalization. We stack $L$ layers to obtain the language-conditioned point cloud features $\hat{V}_t^L \in \mathbb{R}^{N_v/2^{L_e}\times d}$.

**Action decoding.** Since the gripper position $a_{t+1}^{xyz}$ is in a continuous space, previous work [14] discretizing the action space can suffer from quantization error. However, a direct regression approach is often difficult in training leading to unsatisfactory performance [54]. Therefore, we take inspiration from the integral human pose regression [55] to get the best of the both worlds. For the position prediction, we use the PointNext decoder [17] with residual connection and upsampling to generate a heatmap over the input point cloud $H \in \mathbb{R}^{N_v\times1}$ and an offset at each point $\Delta \in \mathbb{R}^{N_v\times3}$. Albeit the point cloud only contains points of visible objects, the action position can cover the whole workspace. Thus we predict an offset to shift each point. The final predicted position is as follows:

$$\hat{a}_{t+1}^{xyz} = \sum_{i=1}^{N_v} H_i(p_i + \Delta_i). \tag{5}$$

It allows continuous output and the underlying heatmap representation makes it easy to train. The rotation and open state are more discrete in nature, thus we directly predict them as follows:

$$\hat{a}_{t+1}^q, \hat{a}_{t+1}^o = \text{MLP}([\text{MaxPool}(V_t^{L_e}); \text{MaxPool}(\hat{V}_t^L)]). \tag{6}$$

### 3.2.3 Training Objective

We use behavioral cloning for model training. Given the dataset $\mathsf{D} = \{(X_i, \tau_i)\}_{i=1}^N$ with $N$ pairs of successful demonstration $\tau_i$ and the corresponding instruction $X_i$, where $\tau_i$ is composed of a sequence of $T$ key steps with visual observations $\{o_t\}_{t=1}^T$, and actions $\{a_t\}_{t=1}^T$. The training objective is to minimize the following loss, including a mean square error (MSE) on the gripper's position and rotation and a binary cross-entropy (BCE) loss on the open state classification:

$$\mathscr{L} = \frac{1}{|NT|}\sum_{\tau\in\mathsf{D}}\left[\sum_{t=1}^T \text{MSE}\left(\hat{a}_t^{xyz}, a_t^{xyz}\right) + \text{MSE}\left(\hat{a}_t^q, a_t^q\right) + \text{BCE}\left(\hat{a}_t^o, a_t^o\right)\right]. \tag{7}$$

**Table 1:** Comparison of point cloud representations on the single-task single-variation evaluation.

| RGB | Normal | Height | Pick & Lift | Pick-Up Cup | Put Knife | Put Money | Push Button | Reach Target | Slide Block | Stack Wine | Take Money | Take Umbrella | Avg. |
|---|---|---|---|---|---|---|---|---|---|---|---|---|---|
| × | × | × | 26.2 | 44.0 | 81.1 | 95.9 | 99.6 | 27.8 | 89.3 | 91.0 | 70.3 | 95.3 | 72.1 $_{\pm 4.4}$ |
| ✓ | × | × | 97.9 | 94.7 | 79.5 | 95.8 | 100.0 | 100.0 | 91.0 | 91.1 | 65.9 | 97.3 | 91.3 $_{\pm 1.6}$ |
| ✓ | ✓ | × | 94.9 | 94.2 | 77.1 | 95.9 | 90.3 | 100.0 | 93.1 | 94.1 | 69.4 | 94.4 | 90.3 $_{\pm 3.1}$ |
| ✓ | × | ✓ | 96.2 | 94.3 | 82.6 | 95.3 | 100.0 | 99.9 | 91.5 | 90.3 | 67.5 | 97.7 | 91.5 $_{\pm 1.4}$ |
| ✓ | ✓ | ✓ | 96.7 | 91.9 | 82.5 | 96.1 | 99.9 | 99.9 | 93.5 | 94.1 | 68.6 | 97.5 | **92.1** $_{\pm 0.4}$ |

## 4 Experiments

### 4.1 Experimental Setup

**Evaluation setup.** We consider three evaluation setups on RLBench [19]. The *single-task single-variation [3]* consists of N tasks with one variation per task. We generate 100 demonstrations for each task variation. Separate models are trained for each task. The *multi-task single-variation* uses the same N tasks as the single-task single-variation setup, but a unified model is trained to solve all the tasks. We consider N={10, 74} following the previous work [3], where 74 is the maximum number of tasks that can be successfully executed in RLBench. The *multi-task multi-variation* includes 18 tasks where each task has multiple variations, resulting in 249 variations in total [14]. We use the same 100 demonstrations for each task as [14]. A single model is trained for all the task variations, which is more challenging due to the larger set of task variations and fewer training data per task variation. More details of the three setups are presented in Section A in the appendix.

**Evaluation metrics.** We use the task success rate (SR) to measure the performance which is 1 for complete success and 0 for failure per episode with no partial credits. For the first two setups, we evaluate on 500 unseen episodes per task ($500 \times 10 = 5000$ evaluation episodes in total) following [3]. We run experiments three times with three seeds and report the mean and standard deviation. For the last setup, we use the released dataset from [14] and evaluate on 25 episodes per task as [14] with a total number of $25 \times 18 = 450$ evaluation episodes.

**Implementation details.** For the model architecture, we use the PointNext-S model [17] with $L_e = 4$ SA blocks and compare $d_e = \{32, 64\}$. We initialize it with the weights trained on ShapeNet. The transformer layers have $d = 512$ and we compare $L = \{1, 2, 4\}$. We use the AdamW optimizer with a learning rate of $5 \times 10^{-4}$ and a batch size of 8 demonstrations. We train for 20K, 200K and 600K for the single-task single-variation, multi-task single-variation and multi-task multi-variation setups respectively. We use a single NVIDIA V100 GPU for training except in the multi-task multi-variation setup where we use 4 GPUs for acceleration. Our model is efficient to train, taking about 1 hour for single-task training and 30 hours to train for multi-task multi-variation.

### 4.2 Ablations

We study the impact of point cloud inputs and model architectures. Unless stated otherwise, we use a point cloud representation combining XYZ, RGB, normal and height, remove background and table points, and normalize them with the gripper position as the origin. The model uses $d_e = 32, L = 1$. The ablations are performed in the single-task single-variation setup with 10 tasks.

**Point cloud representation.** Table 1 compares point cloud input representations. The vanilla point cloud with only XYZ performs worst on average. The RGB color plays an important role for tasks requiring to distinguish colors such as "Pick-Up Cup" where the goal is to pick up the red cup amongst differently colored cups. The improvement from normal of the point is less stable, which might result from noisy normal estimation. The height of point relative to the table can also slightly improve the performance.

**Table 2:** Comparison of point cloud processing on the single-task single-variation evaluation.

| Coord origin | Remove Table | Remove Background | Avg. |
|---|---|---|---|
| Center | ✓ | ✓ | 92.1 $_{\pm 2.0}$ |
| Gripper | × | × | 81.6 $_{\pm 3.2}$ |
| | × | ✓ | 89.9 $_{\pm 2.8}$ |
| | ✓ | ✓ | **92.1** $_{\pm 0.4}$ |

**Table 5:** Comparison with state-of-the-art methods on single-task single-variation, multi-task single-variation and multi-task multi-variation setups.

| | 10 Tasks | | 74 Tasks | | 18 Tasks |
| | Single-task | Multi-task | Single-task | Multi-task | Multi-task Multi-var |
|---|---|---|---|---|---|
| Auto-$\lambda$ [2] | 55.0 | 69.3 | - | - | - |
| PerAct [14] | - | - | - | - | 42.7 |
| Hiveformer [3] | 88.4 | 83.3 | 66.1 | 49.2 | 45.3 |
| PolarNet (Ours) | **92.1** | **89.8** | **69.0** | **60.3** | 46.4 |

**Table 3:** Comparison of camera views on the single-task single-variation evaluation.

| Left | Right | Wrist $\parallel$ | Avg. |
|---|---|---|---|
| ✓ | × | × | 37.6 $_{\pm 4.8}$ |
| × | ✓ | × | 48.0 $_{\pm 4.5}$ |
| × | × | ✓ | 35.0 $_{\pm 5.5}$ |
| ✓ | ✓ | × | 67.0 $_{\pm 4.7}$ |
| ✓ | × | ✓ | 80.2 $_{\pm 3.0}$ |
| × | ✓ | ✓ | 76.6 $_{\pm 5.6}$ |
| ✓ | ✓ | ✓ | **92.1** $_{\pm 0.4}$ |

**Point cloud processing.** We compare two coordinate frames to normalize the point cloud, using the gripper position or the points center as the origin. As shown in Table 2, the two ways perform similarly while using the gripper position is more robust with lower standard deviation. More important in point cloud processing is to remove irrelevant points. Since we have a limited capacity to sample points, the irrelevant points make the interesting points sparser and harm the performance. Results on the multi-task setup also shows the same trend as in Table 12 in the appendix.

**Multi-view cameras.** Table 3 analyzes the contributions from multi-view cameras for point cloud construction. A single camera alone is insufficient to accomplish a task due to occlusions, decreasing the performance by more than 44%. The wrist camera performs worst among the three views. However, it is more complementary to the other two cameras, with more than 10% boost compared to the combination of left and right shoulder cameras. Using all the three cameras achieves the best performance with a significant margin (+11.9%, +15.5%, + 25.1% depending on the resp. two cameras).

**Model capacity.** We explore the influence of model capacities to the task performance in single- and multi-task setups in Table 4. In the single-task setup, a small network with $d_e = 32$ performs slightly better than a larger network with $d_e = 64$. We suspect that the larger model may require more demonstrations to train. However, the same model capacity does not perform well in the multi-task setup. It is essential to increase the model size to learn multiple manipulation tasks together. We also notice that the best model in multi-task still underperforms that in single-task setup by 2.3%, indicating that a better multi-task learning strategy is in need. In the following, we use $d_e = 32, L = 1$ for single-task setting and $d_e = 64, L = 2$ for the multi-task settings.

**Table 4:** Comparison of model capacity on single-task and multi-task (MT) setups.

| MT | $d_e$ | $L \parallel$ | Avg. |
|---|---|---|---|
| × | 32 | 1 | **92.1** $_{\pm 0.4}$ |
| | 64 | 1 | 91.0 $_{\pm 0.9}$ |
| ✓ | 32 | 1 | 77.4 $_{\pm 8.6}$ |
| | 64 | 1 | 89.3 $_{\pm 2.0}$ |
| | 64 | 2 | **89.8** $_{\pm 1.5}$ |
| | 64 | 4 | 86.3 $_{\pm 4.8}$ |

### 4.3 Comparison with State of the Art

**Evaluation on three setups.** We compare our PolarNet with state-of-the-art approaches on single-task single-variation, multi-task single-variation and multi-task multi-variation setups in Table 5. More detailed results on each task of the three setups are provided in Table 9, 10, and 11 respectively in the appendix. The compared methods include 2D image based models such as Auto-$\lambda$ [2] and Hiveformer [3] and 3D voxel based model PerAct [14]. Our PolarNet consistently outperforms prior models on all the evaluation setups. It achieves considerable improvement with more tasks in the multi-task setup (+11.1), demonstrating the advantages of 3D point clouds to solve multiple tasks simultaneously. In addition, our model is more computationally efficient compared to previous 3D based model PerAct [14] which requires 16 days for training with 8 V100 GPUs. We only use 4 V100 GPUs to train the same amount of iterations in 30 hours, which is roughly 13 times faster. We analyze the success and failure cases in Section D in the appendix.

**Robustness to viewpoint perturbation.** We perform a new evaluation on robustness of viewpoint variances to demonstrate the advantages of the point cloud model compared to 2D image based methods. In the evaluation, we change the camera positions and rotations to obtain new RGB-D images.

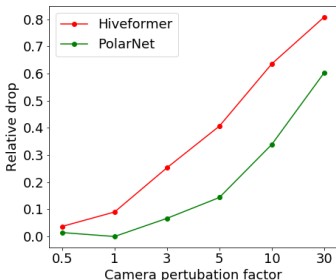

**Figure 3:** Comparison of different models on the robustness to camera perturbation, where the lower the better.

We only apply the change to the two side cameras but keep the wrist camera the same, because in real world experiments the side cameras are easy to place in different locations but the wrist camera is always attached to the robotic hand. We denote the perturbation factor $f$ as randomly selecting $\pm f$ cm shift and $\pm 5f$ degrees for the modified camera pose. We directly run the PolarNet and Hiveformer models trained on the original cameras in the multi-task setup of 10 tasks with the new camera poses. The results are presented in Figure 3. The performance of PolarNet drops much less than Hiveformer for all the perturbation factors, demonstrating the robustness of our PolarNet on camera perturbation. The detailed results on each of the 10 tasks are presented in Figure 9 in the appendix.

## 4.4 Real-robot Experiments

We further evaluate PolarNet with real visual sensors on a real robot. We use a 6-DoF UR5 robotic arm and adopt 7 real-world tasks. For each task, we collect 20 human demonstrations and fine-tune a language-conditioned multi-task model trained on RLBench on the collected data. More details are presented in Section E in the appendix. Table 6 presents the real-robot results. We report the success rate on 10 episodes per task. Our PolarNet achieves 60% success rate on average for the 7 real-world tasks. We summarize two prevalent reasons of failures in PolarNet. First, the model confuses the target object such "strawberry" and "apple" since the sparse point cloud may not provide sufficient semantic information. Secondly, the predicted grasping pose lacks precision. We conjecture that this is due to partial views and the multimodal nature of the target action distribution. To tackle these limitations, further research could be conducted on pretraining semantic-aware 3D representations [56], combining it with pretrained 2D models, and improving the complexity of action policy. Please see our project website [57] for examples of the policy execution in the real world.

**Table 6:** Performance of PolarNet on 7 real-world tasks.

| Task | PolarNet |
| --- | --- |
| Stack cup | 8/10 |
| Put fruit in box | 8/10 |
| Put plate on table | 3/10 |
| Open drawer | 9/10 |
| Put item in drawer | 4/10 |
| Put item in cabinet | 4/10 |
| Hang mug | 6/10 |
| Average | 6/10 |

## 5 Conclusion

This work addresses language-guided robotic manipulation using 3D point clouds. The proposed PolarNet employs carefully designed point cloud inputs, an efficient point cloud encoder, and a multimodal transformer to predict 7-DoF actions for language-conditioned manipulation. We find that it is critical to use point color with colors, filter irrelevant points and merge multiple views. Extensive experiments on the RLBench benchmark demonstrate that PolarNet outperforms state-of-the-art models on a variety of tasks, from single-task single-variation to multi-task multi-variation. The PolarNet also achieves promising results on a real robot to solve multiple tasks.

**Limitations:** Our multi-task model still does not perform as well as the best single-task model, requiring more advanced multi-task learning algorithms. Additionally, while our policy can perform multiple tasks, we have not studied the generalization to new scenes, objects, and tasks.

## Acknowledgments

This work was partially supported by the HPC resources from GENCI-IDRIS (Grant 20XX-AD011012122). It was funded in part by the French government under management of Agence Nationale de la Recherche as part of the "Investissements d'avenir" program, reference ANR19-P3IA-0001 (PRAIRIE 3IA Institute), the ANR project VideoPredict (ANR-21-FAI1-0002-01) and by Louis Vuitton ENS Chair on Artificial Intelligence.

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

# A  RLBench Tasks

We consider RLBench tasks for three different learning setups: single-task single-variation, multi-task single-variation and multi-task multi-variation.

In the first two setups, we adopt a 10-task setting and a 74-task setting following [3]. Figure 4 depicts the 10 tasks and the corresponding instructions. Table 7 presents the details of all 74 tasks that can be successfully executed in RLBench. The 74 tasks are manually grouped into 9 categories by [3]. We use the original RLBench code[2] to collect training data and run evaluation.

In the multi-task multi-variation setup, we use 18 tasks with 249 variations following [14]. Table 8 lists the task names, variation type, the number of variations and examples of instructions. We use RLBench codebase[3] from [14] which modified some tasks to have more variations. To match the training and evaluation setup from [14] we use the datasets provided in PerAct code repository.

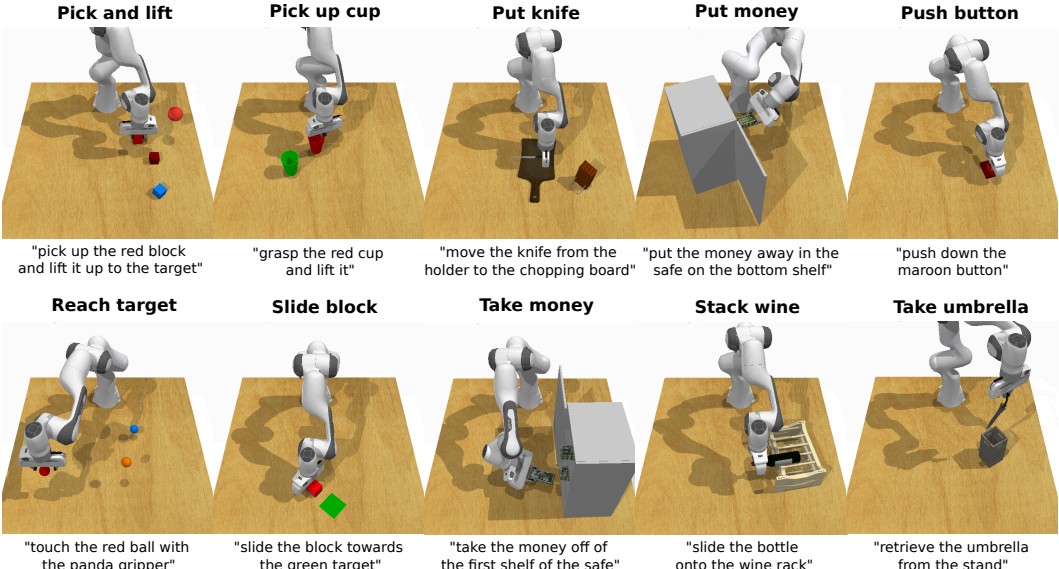

**Figure 4:** Examples of the selected 10 tasks with corresponding instructions in RLBench.

# B  Model Details

**Keysteps actions.** For robotic control, we use macro steps [34] – key turning points in action trajectories where the gripper changes its state (open/close) or velocities of joints are close to zero. In this way, the sequence length of an episode is significantly reduced from hundreds of small steps to typically less than 10 macro steps.

**Point removal in point cloud preprocessing.** In Figure 5, we show the raw point cloud generated from multi-view cameras, the point cloud with background and table removal. As the background such as wall and floor are geometrically far away from the robot workspace and the table is in a fixed height, we could simply pre-define a 3D bounding box to select relevant points from the raw point cloud. This is more efficient than performing object segmentation to obtain points of objects.

**PolarNet architecture.** Figure 6 presents a detailed architecture of our PolarNet, which utilizes PointNext [17] backbone for point cloud encoding, multimodal transformers for vision-and-language interaction and PointNext decoder for action prediction.

---

[2]https://github.com/stepjam/RLBench
[3]https://github.dev/MohitShridhar/RLBench/blob/peract/

**Table 7:** Details of the selected 74 tasks in RLBench.

| Group | #Tasks | Task Names |
|---|---|---|
| Planning | 9 | basketball_in_hoop, change_channel, meat_off_grill, meat_on_grill, push_buttons, put_rubbish_in_bin, stack_wine, tower3, tv_on |
| Tools | 11 | hang_frame_on_hanger, move_hanger, place_hanger_on_rack, reach_and_drag, scoop_with_spatula, screw_nail, slide_block_to_target, sweep_to_dustpan, take_frame_off_hanger, take_plate_off_colored_dish_rack, water_plants |
| Long Term | 4 | slide_cabinet_open_and_place_cups, stack_blocks, take_shoes_out_of_box, wipe_desk |
| Rotation-invariant | 6 | lamp_off, lamp_on, pick_and_lift, push_button, reach_target, take_lid_off_saucepan |
| Motion planner | 9 | close_box, close_drawer, close_laptop_lid, open_box, open_drawer, phone_on_base, put_books_on_bookshelf, toilet_seat_down, toilet_seat_up |
| Multimodal | 5 | beat_the_buzz, lift_numbered_block, pick_up_cup, stack_cups, turn_tap |
| Precision 12 | 12 | insert_onto_square_peg, insert_usb_in_computer, pick_and_lift_small, place_shape_in_shape_sorter, play_jenga, put_knife_on_chopping_board, put_umbrella_in_umbrella_stand, setup_checkers, straighten_rope, take_toilet_roll_off_stand, take_umbrella_out_of_umbrella_stand, take_usb_out_of_computer |
| Screw | 4 | change_clock, open_window, open_wine_bottle, turn_oven_on |
| Visual occlusion | 14 | close_door, close_fridge, close_grill, close_microwave, open_door, open_fridge, open_grill, open_microwave, open_oven, plug_charger_in_power_supply, press_switch, put_money_in_safe, take_money_out_safe, unplug_charger |

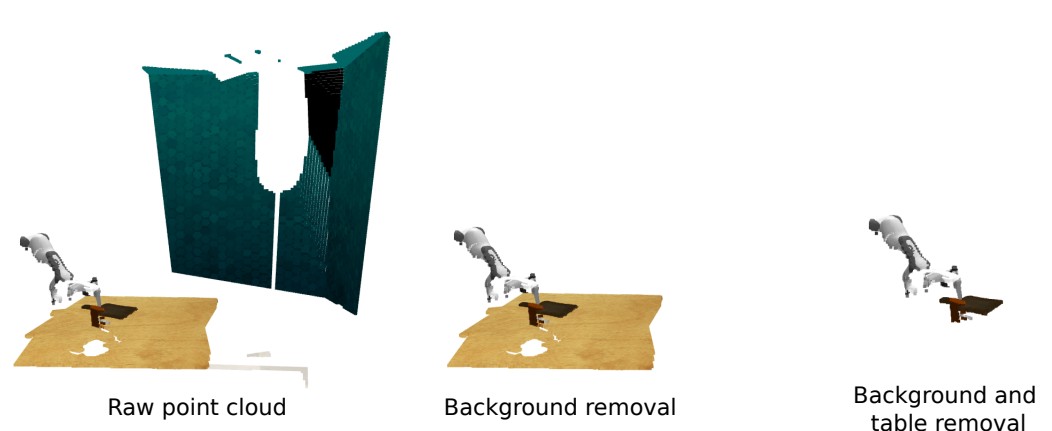

Raw point cloud     Background removal     Background and table removal

**Figure 5: Illustration of point cloud processing.** We represent the raw point cloud, point cloud with background removal and point cloud with background and table removal for put knife task.

**Table 8: Details of the 18 tasks with 249 variations in the multi-task multi-variation setup.**

| Task | Var. type | # vars. | Example of language template |
|---|---|---|---|
| open drawer | placement | 3 | "open the __ drawer" |
| slide block | color | 4 | "slide the block to __ target" |
| sweep to dustpan | size | 2 | "sweep dirt to the __ dustpan" |
| meat off grill | category | 2 | "take the __ off the grill" |
| turn tap | placement | 2 | "turn __ tap" |
| put in drawer | placement | 3 | "put the item in the __ drawer" |
| close jar | color | 20 | "close the __ jar" |
| drag stick | color | 20 | "use the stick to drag the cube onto the __ target" |
| stack blocks | color, count | 60 | "stack __ __ blocks" |
| screw bulb | color | 20 | "screw in the __ light bulb" |
| put in safe | placement | 3 | "put the money away in the safe on the __ shelf" |
| place wine | placement | 3 | "stack the wine bottle to the __ of the rack" |
| put in cupboard | category | 9 | "put the __ in the cupboard" |
| sort shape | shape | 5 | "put the __ in the shape sorter" |
| push buttons | color | 50 | "push the __ button, [then the __ button]" |
| insert peg | color | 20 | "put the ring on the __ spoke" |
| stack cups | color | 20 | "stack the other cups on top of the __ cup" |
| place cups | count | 3 | "place __ cups on the cup holder" |

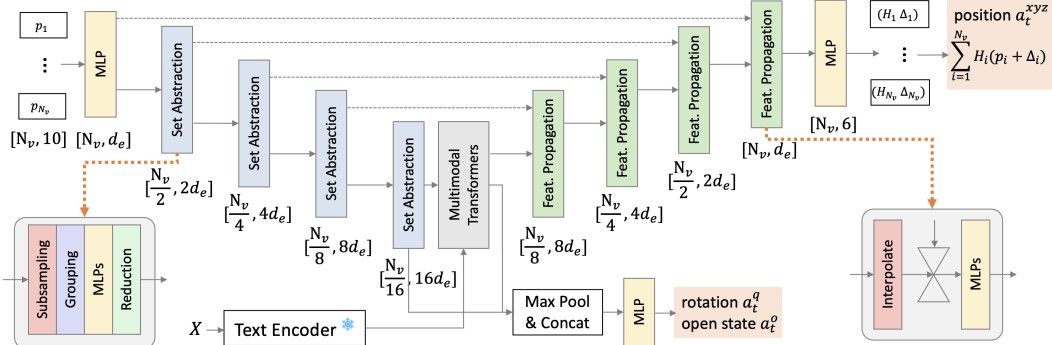

**Figure 6:** Detailed architecture of PolarNet for language guided robotic manipulation. The model takes the input of point cloud $\{p_1, ..., p_{N_v}\}$ and instruction $X$, where $N_v$ is the number of points and $d_e$ is the dimensionality of the feature. It outputs the position $a_t^{xyz}$, rotation $a_t^q$ and open state $a_t^o$ of the gripper for the next keystep. To compute the position $a_t^{xyz}$, it first predicts a heapmap $H_i$ and a position offset $\Delta_i$ for each point and then compute the integral.

**Hiveformer [3] re-implementation.** We implemented Hiveformer [3] by ourselves and achieve similar performance as they reported in the paper on the multi-task single-variation setting. To use Hiveformer on the challenging multi-task setups, we enlarge the model capacity by doubling the hidden size and train the same number of iterations with PolarNet, which leads to better performance than using the hyper-parameters as in the original paper.

# C   Quantitative Results in Simulation

**Results on 10 tasks.** Table 9 presents the detailed results for 10 tasks in the single-task and multi-task single-variation setups. We compare our model with ARM [34], Auto-$\lambda$ [2] and Hiveformer [3]. All the three methods are based on multi-view 2D RGB-D images for action prediction. Our point cloud based model obtains better and more stable results compared to the previous best model Hiveformer ($92.1\pm0.4$ vs. $88.4\pm4.9$) in the single-task setup, demonstrating the effectiveness of the 3D representations. The improvements are consistent across tasks except for "Take Money". We observe that the main problem on the "Take Money" task is due to the imperfect motion planner. As

**Table 9:** Comparison with state-of-the-art methods on 10 tasks in single-task and multi-task setups. All the methods are trained on 100 demonstrations per task. We report the success rate (%) of our method on 500 episodes per task averaged over 3 seeds.

| | Pick & Lift | Pick-Up Cup | Put Knife | Put Money | Push Button | Reach Target | Slide Block | Stack Wine | Take Money | Take Umbrella | Avg. |
|---|---|---|---|---|---|---|---|---|---|---|---|
| *Single-task single-variation* | | | | | | | | | | | |
| ARM [34] | 70 | 80 | - | - | - | 100 | - | 70 | - | 70 | - |
| Auto-$\lambda$ [2] | 82 | 72 | 36 | 31 | 95 | 100 | 36 | 23 | 38 | 37 | 55.0 |
| Hiveformer [3] | 92.2 | 77.1 | 69.7 | 96.2 | 99.6 | 100.0 | 95.4 | 81.9 | 82.1 | 90.1 | 88.4 |
| PolarNet (Ours) | 96.7 | 91.9 | 82.5 | 96.1 | 99.9 | 99.9 | 93.5 | 94.1 | 68.6 | 97.5 | **92.1** |
| *Multi-task single-variation* | | | | | | | | | | | |
| Auto-$\lambda$ [2] | 87 | 78 | 31 | 62 | 95 | 100 | 77 | 19 | 64 | 80 | 69.3 |
| Hiveformer [3] | 88.9 | 92.9 | 75.3 | 58.2 | 100.0 | 100.0 | 78.7 | 71.2 | 79.1 | 89.2 | 83.3 |
| PolarNet (Ours) | 97.8 | 86.0 | 80.5 | 94.1 | 99.6 | 100.0 | 93.4 | 80.5 | 68.1 | 97.8 | **89.8** |

**Table 10:** Comparison with state-of-the-art methods on 74 tasks in single-task and multi-task setups. We evaluate 500 episodes per task. Besides reporting averaged success rate (%), we group tasks into 9 categories and report the averaged success rate per group. († denotes our own implementation.)

| | Planning | Tools | Long Term | Rot. Invar. | Motion Planning | Screw | Multi Modal | Precision | Visual Occlusion | Avg. tasks |
|---|---|---|---|---|---|---|---|---|---|---|
| # of tasks | 9 | 11 | 4 | 6 | 9 | 4 | 5 | 12 | 14 | 74 |
| *Single-task single-variation* | | | | | | | | | | |
| Hiveformer [3][†] | 81.1 | 60.6 | 7.6 | 95.6 | 75.8 | 82.4 | 64.0 | 53.6 | 65.4 | 66.1 |
| PolarNet (Ours) | 90.5 | 66.8 | 5.5 | 96.4 | 75.6 | 73.1 | 61.7 | 59.0 | 69.1 | **69.0** |
| *Multi-task single-variation* | | | | | | | | | | |
| Hiveformer [3][†] | 65.9 | 27.7 | 3.4 | 78.9 | 72.8 | 66.1 | 48.5 | 35.5 | 47.8 | 49.2 |
| PolarNet (Ours) | 75.5 | 46.3 | 6.9 | 87.8 | 83.3 | 63.0 | 54.5 | 49.0 | 61.0 | **60.3** |

shown in Figure 7, when the check is placed inside of the safe, the robotic arm often collides with the safe door even if the predicted action is reasonable. Because of this, the groundtruth keystep action only achieves 89% success rate. In the multi-task setup, our model also surpasses the state-of-the-art methods with 6.5% absolute gains compared to Hiveformer [3].

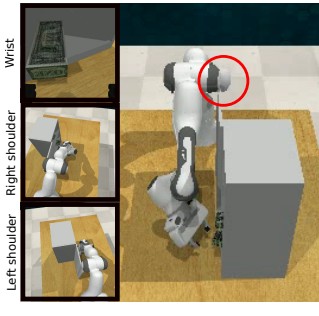

**Figure 7:** Failure case of "Take Money" task due to imperfect motion planner.

**Results on 74 tasks.** In Table 10, we provide the comparison with Hiveformer [3] on 74 tasks in the single-task and multi-task single-variation setup. We achieve 2.9% absolute gains on average for the single-task learning. The improvement becomes more considerable in the multi-task setup (+11.1), demonstrating the advantages of 3D point clouds to solve multiple tasks simultaneously. Our 3D model outperforms the 2D model on tasks that require planning, use tools, need precise control, and have visual occlusion.

**Results on 18 tasks in the multi-task multi-variation setup.** In Table 11, we compare with state-of-the-art methods C2FARM-BC [35], PerAct [14] and Hiveformer [3]. For fair comparison, we add an additional front view camera following [14]. Both C2FARM-BC and PerAct utilize 3D voxels as inputs to train the manipulation policy. Our proposed point cloud based model is more efficient than the two 3D based models. It also improves PerAct by 3.7% on average over 18 tasks. Our 3D-based approach also achieves comparable performance as the state-of-the-art 2D-based approach Hiveformer on the setup which uses histories. It is interesting to further employ histories in our method and unify the 2D and 3D representations in the future work.

**Table 11:** Comparison with state-of-the-art methods on 18 tasks with 249 variations in total. All the methods are trained on 100 demonstrations per task. We report the success rate of our method on 25 episodes per task by default. († denotes our own implementation.)

| | Open Drawer | Slide Block | Sweep to Dustpan | Meat off Grill | Turn Tap | Put in Drawer | Close Jar | Drag Stick | Stack Blocks | |
|---|---|---|---|---|---|---|---|---|---|---|
| C2FARM-BC [35] | 20 | 16 | 0 | 20 | 68 | 4 | 24 | 24 | 0 | |
| PerAct [14] | 80 | 72 | 56 | 84 | 80 | 68 | 60 | 68 | 36 | |
| Hiveformer† [3] | 52 | 64 | 28 | 100 | 80 | 68 | 52 | 76 | 8 | |
| PolarNet (Ours) | 84 | 56 | 52 | 100 | 80 | 32 | 36 | 92 | 4 | |

| | Screw Bulb | Put in Safe | Place Wine | Put in Cupboard | Sort Shape | Push Buttons | Insert Peg | Stack Cups | Place Cups | Avg. |
|---|---|---|---|---|---|---|---|---|---|---|
| C2FARM-BC [35] | 8 | 12 | 8 | 0 | 8 | 72 | 4 | 0 | 0 | 16 |
| PerAct [14] | 24 | 44 | 12 | 16 | 20 | 48 | 0 | 0 | 0 | 42.7 |
| Hiveformer† [3] | 8 | 76 | 80 | 32 | 8 | 84 | 0 | 0 | 0 | 45.3 |
| PolarNet (Ours) | 44 | 84 | 40 | 12 | 12 | 96 | 4 | 8 | 0 | **46.4** |

**Table 12:** Comparison of point removal on the single-task and multi-task setups with 10 tasks.

| Remove Table | Remove Background | Pick & Lift | Pick-Up Cup | Put Knife | Put Money | Push Button | Reach Target | Slide Block | Stack Wine | Take Money | Take Umbrella | Avg. |
|---|---|---|---|---|---|---|---|---|---|---|---|---|
| *Single-task single-variation* | | | | | | | | | | | | |
| × | × | 94.6 | 89.2 | 48.5 | 48.7 | 99.9 | 100.0 | 91.9 | 80.0 | 74.9 | 88.4 | 81.6 |
| × | ✓ | 95.1 | 86.3 | 71.9 | 89.9 | 99.9 | 100.0 | 93.5 | 95.5 | 72.3 | 94.0 | 89.8 |
| ✓ | ✓ | 96.7 | 91.9 | 82.5 | 96.1 | 99.9 | 99.9 | 93.5 | 94.1 | 68.6 | 97.5 | **92.1** |
| *Multi-task single-variation* | | | | | | | | | | | | |
| × | × | 92.2 | 87.0 | 75.4 | 63.6 | 97.2 | 100.0 | 86.6 | 76.4 | 73.2 | 97.0 | 84.9 |
| × | ✓ | 92.0 | 82.6 | 77.2 | 95.4 | 97.4 | 100.0 | 88.2 | 68.2 | 71.6 | 91.6 | 86.4 |
| ✓ | ✓ | 97.8 | 86.0 | 80.5 | 94.1 | 99.6 | 100.0 | 93.4 | 80.5 | 68.1 | 97.8 | **89.8** |

**Data efficiency.** To demonstrate the data efficiency of our model, we reduce the number of demonstrations per task from 100 to 10 in training. We evaluate the multi-task learning setup with 10 tasks and compare our model with the state-of-the-art 2D-based model Hiveformer [3]. The results in Figure 8 show that our 3D-based approach is also data efficient.

**Point removal in PolarNet.** Table 2 has shown that removing irrelevant points in the point cloud processing is critical to the performance. To gain a more comprehensive insight into the improvements, we provide more detailed results of point removal in Table 12 on both single-task and multi-task setups with 10 tasks. The trend in multi-task setup is similar to that in single-task setup, demon-

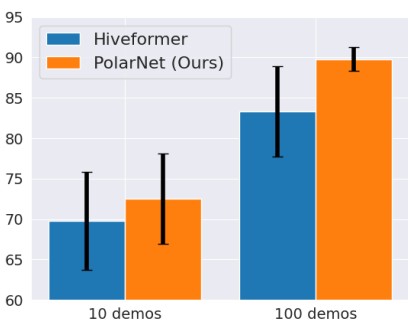

**Figure 8:** Performance of using different numbers of demonstrations per task for training on the multi-task single-variation setup.

strating it is beneficial to remove irrelevant points from the point cloud. We can see that removing background points mainly improves Put Knife, Put Money, Stack Wine and Take Umbrella. In these tasks, the robotic arm rotates a lot at some steps as shown in Figure 5, resulting in more background points. Since the background points are geometrically far away from the workspace of the robot, they drastically affect the normalization of the point cloud and make the PointNext encoding less effective. Removing table points bring more improvements for tasks where the contact areas in the objects are small such as Pick-Up Cup and Put Knife. We also observe that not all the tasks benefit from the point removal. It would be interesting to investigate automatic point removal in the future.

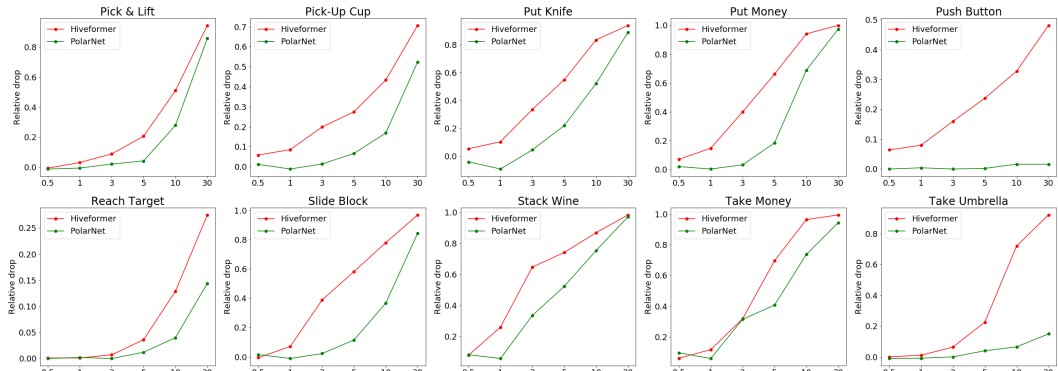

**Figure 9:** Comparison of different models on the robustness to camera perturbation. The x-axis is the level of perturbation. The y-axis is the relative performance drop compared to the results without any perturbations where the lower the better.

**Comparing with Hiveformer with equivalent point removal.** For 2D-based models such as Hiveformer, it is non-trivial to remove the background and table pixels from the image compared to the simple approach we used in 3D-based PolarNet. Segmentation models are required to mask the background and table. Since we are able to obtain the groundtruth segmentation mask in the simulation, we use the groundtruth mask as an upper bound for 2D models. We mask pixels of the background and table as 0 in the image as the input to the model as well as the

**Table 13:** Comparing with an equivalent point removal in Hiveformer on the multi-task setup of 10 tasks.

| Method | Point removal | Avg. |
|---|---|---|
| Hiveformer | × | 82.8 |
| | ✓ | 73.9 |
| PolarNet | × | 84.9 |
| | ✓ | 89.8 |

final predicted heatmap. The results are presented in Table 13. We can see that removing those pixels does not improve the performance. We conjecture that those pixels might be helpful for the 2D-based model to align multi-view images.

**Robustness to viewpoint perturbation.** Figure 9 presents the relative performance drop of different multi-task policies under different camera perturbation factors for each of the 10 task. We can see that our PolarNet is more robust on all the 10 tasks than the 2D-based state-of-the-art model Hiveformer, demonstrating the advantage of 3D-based representations. Particularly, PolarNet achieves stable performance for Push Button, Reach Target and Take Umbrella where the relative performance drops are less than 20% even under the large camera perturbation factor of 30.

## D Qualitative Results in Simulation

**Success cases.** We show successful cases for multi-variation multi-task setting in Figure 10. In drag stick task, the policy is able to use a stick to drag the cube to the colored target given by the language instruction. We observed that policy can adapt to the cube trajectory drift to complete the task. Policy is also able to generalize to placement variation such as place wine task. We provide more complex and additional success cases on the supplementary video.

**Failure cases.** In Figure 11 we show different failure cases of our PolarNet. We can observe how the policy fails to insert the ring on the blue spoke due to imprecise position prediction in insert peg task. This task requires fine-grained manipulation which is difficult to have when controlling the robot with keysteps. Place cups is a task that can be solved by placing the cups on the tree in any order. Due to the multimodality nature of this task, we observe that the robot fails to grasp a cup going in between two cups. Finally, in some tasks such as sort shape, the motion planner fails causing the gripper to collide with other objects in the scene failing to complete the task.

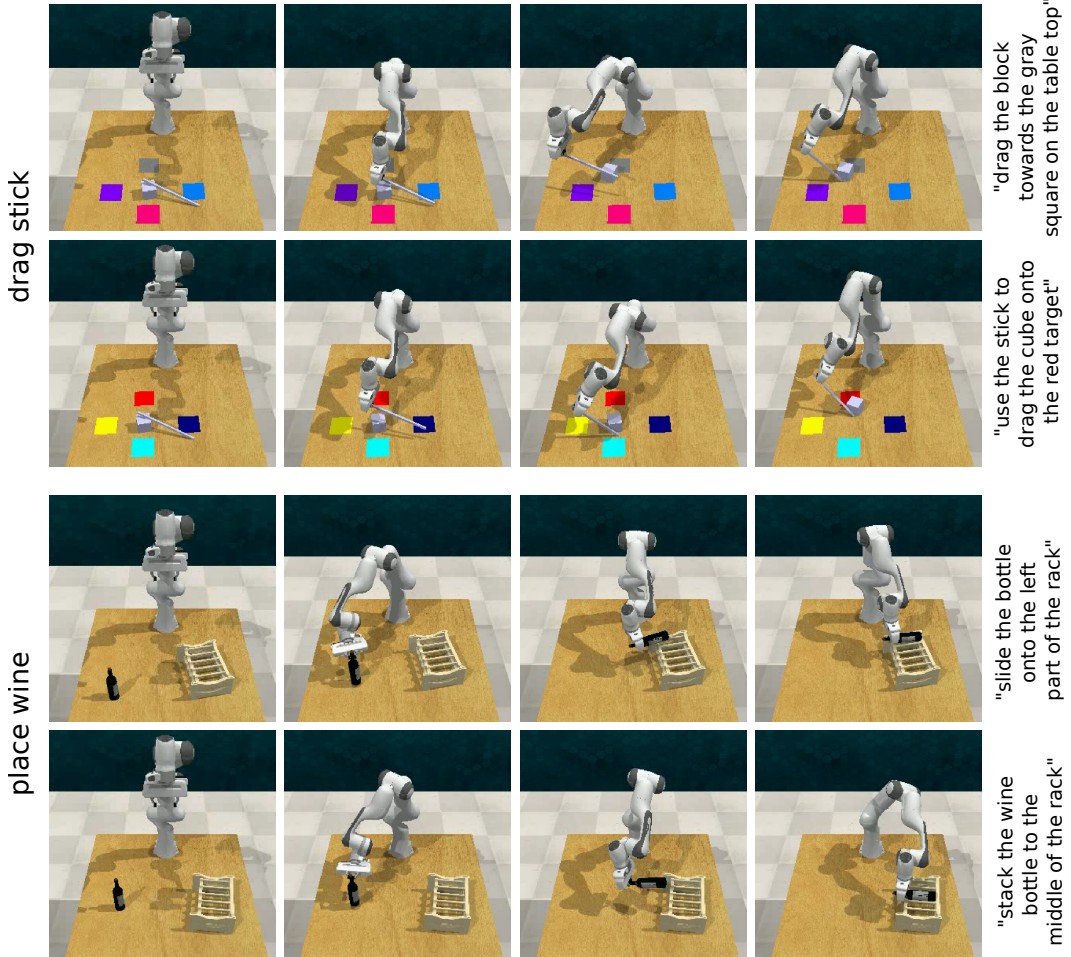

**Figure 10: PolarNet success cases on multi-task multi-variation setting.** We show policy running examples of the drag stick and place wine tasks.

## E  Real-robot Experiments

**Hardware setup.** We conduct real-robot experiments on a 6-DoF UR5 robotic arm equipped with a RG6 gripper. Two Intel RealSense D435 RGB-D cameras are set on the front and lateral sides of the scene as illustrated in Figure 12. To be noted, our gripper is larger than the one used in RLBench, which makes it more challenging to predict precise gripper positions. Furthermore, our robotic arm's base is situated besides a column instead of directly on the table, resulting in additional difficulties in motion planning. For example, the robot needs to perform substantial rotations to prevent collisions with the column.

**Tasks and data collection.** We adopt 7 real-world tasks including *stack cup*, *put fruit in box*, *open drawer*, *hang mug*, *put item in drawer*, *put item in cabinet* and *put plate on table* as illustrated in Figure 13. We randomly place the target objects and multiple distractors on the table for each episode of a task. Due to a large gap between simulated and real environments, we collect a few real-robot demonstrations to finetune the policy trained in the simulator. Specifically, we use a joystick controller to manually control the robot. We define the keysteps for the tasks and save RGB-D images and proprioceptive information of the gripper at each keystep. Finally, we collected 20 demonstrations per task.

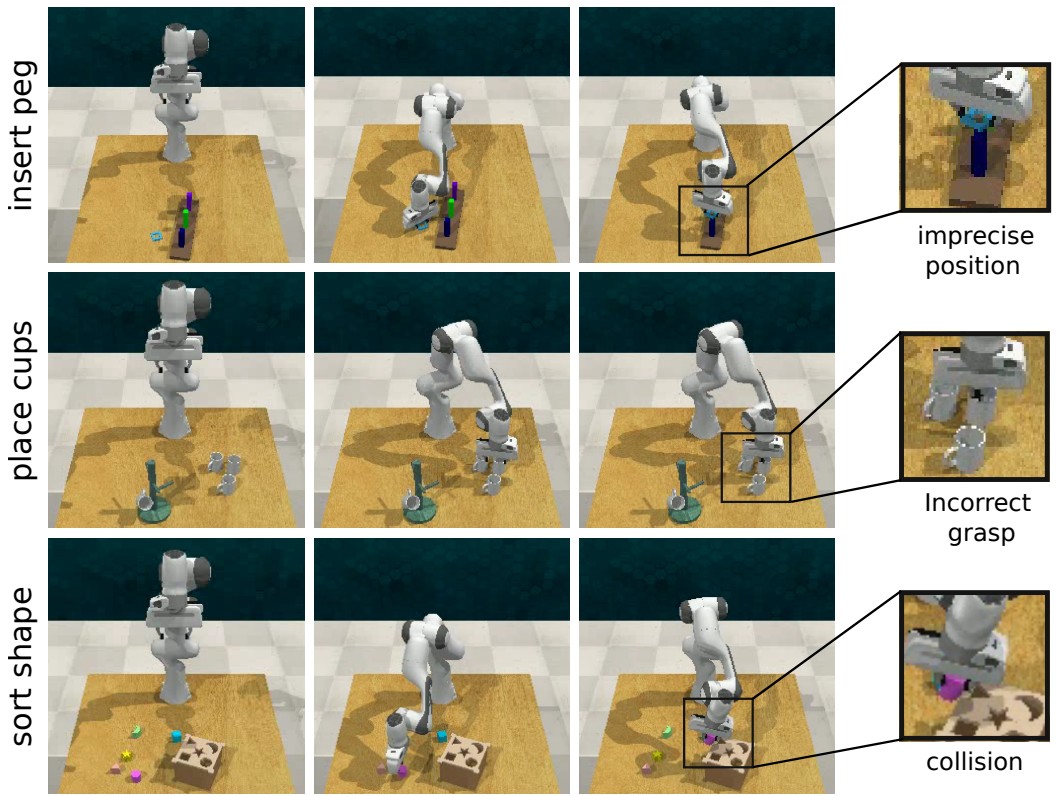

**Figure 11: PolarNet failure cases.** We illustrate failure cases for insert peg, place cups and sort shape.

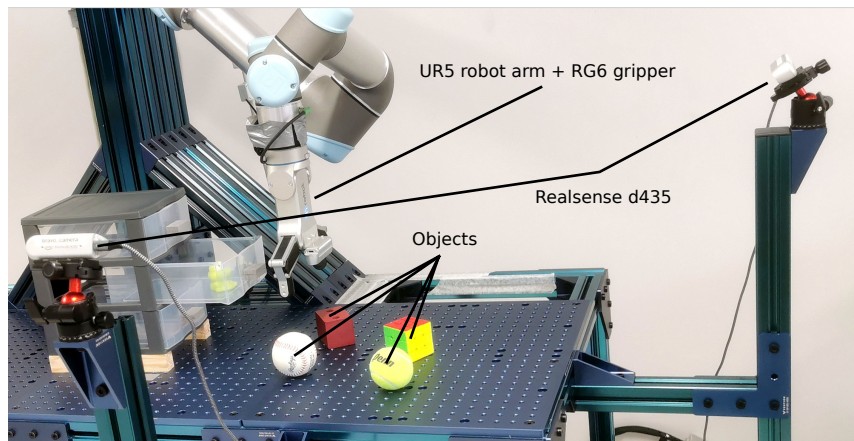

**Figure 12: Robotic setup.** Our setup includes two RealSense D435 cameras, the objects and the UR5 robotics arm equipped with a RG6 gripper.

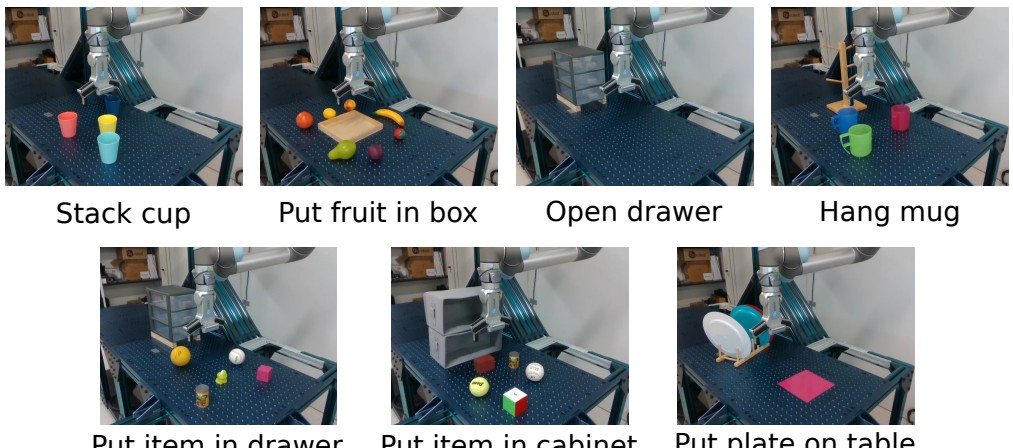

| Stack cup | Put fruit in box | Open drawer | Hang mug |

| Put item in drawer | Put item in cabinet | Put plate on table |

**Figure 13:** Illustration of the 7 real-world tasks.

**Training and execution.** We finetune the PolarNet trained in the multi-task setup in RLBench with the collected real-robot data. Though the depth images are noisy in the real-world setup, we do not perform other preprocessing algorithms to improve the obtained point cloud except the outlier removing[4]. We train on 1 NVIDIA V100 GPU for 100K iterations which takes around 9 hours. During evaluation, we simply use the last checkpoint from training. We use a NVIDIA GeForce GTX 1080 Ti GPU for inference. To move between keysteps, we use MoveIt motion planner framework.

---

[4]http://www.open3d.org/docs/latest/tutorial/Advanced/pointcloud_outlier_removal.html

