# OpenReview forum: "PolarNet: 3D Point Clouds for Language-Guided Robotic Manipulation"
_robot-learning.org/CoRL/2023/Conference — CoRL 2023 Poster_

### Official Review · Reviewer_dUV7 · 2023-07-14

**Confidence:** 3
**Originality:** Good
**Technical Quality:** Very Good
**Clarity Of Presentation:** Good
**Impact:** 4

**Recommendation:**

Weak Accept: I recommend accepting the paper, but will not argue for my recommendation if the majority of other reviewers have a different opinion.

**Review:**

Originality:

The problem and the method are both relatively new. It proposed to use point clouds as the feature space for the manipulation problem with RGBD inputs. It incorporated representative solutions in point cloud learning and large language models and leveraged transformers to fuse cross-modal features.

Quality:

The proposed technique is presented clearly, correct, and reasonable. The presented work is a complete piece as a conference paper. The experiment section is thorough. It not only compared with existing work, but detailed ablation study in terms of point cloud normalization, combination of input information, point cloud preprocessing (removing irrelevant points), number of camera inputs, and model size are presented. One shortcoming is the lack of real-world experiments.

Overall, while the components are mature results from several relevant fields, this paper presented an effective way of putting them together to fulfill the task, and the details in the realization are also validated through experiments.

Clarity:

The motivation, background, methodology, and experiments are presented mostly clearly. I suggest the author to more clearly explain what the "variations" involve, when mentioning "multi-task multi-variation", etc.

Significance:

The proposed framework for language-guided RGBD-based manipulation is valuable and researchers are likely to build improved solutions based on this method in the future.

Relevance:

The problem addressed in this paper is of significant practical value and could attract the interest of CoRL's audience.

Limitations:
The paper mentioned that this work did not explore the generalization of out-of-distribution scenes, objects, and tasks, which I agree. Also, the method requires a lot of supportive information to work properly, including the removal of irrelevant points, known camera poses, and dense depth images. In my understanding, this is a prototype work incorporating language models and point clouds for manipulation.

Strengths:

1. Proposed to combine LLM and point cloud representation for manipulation.

2. Obtained superior performance in experiments.

Weaknesses:

1. No real-world experiments.

2. Cannot generalize to unseen tasks or scenes despite the application of large language models.

**Quality Of The Limitations Section:**

Limitations are addressed clearly

**Questions For Rebuttal:**

1. Could you explain more on the variations involved in the experiments?

2. What are the possible ways to enable better generalization?

3. What are the possible ways to remove some requirements on the input other than the images?

**Robotics Focus:**

Highly relevant to robotics but no hardware experiments

**Summary Of Paper:**

This paper proposed a framework for language-guided robot manipulation leveraging 3D point cloud representations and the text embeddings from large pretrained language models. Point-cloud representation is the highlight of this work, as it benefits efficiency, sensor fusion, and accurate 3D localization.

**Summary Of Recommendation:**

This work is a pioneer in combining language models and RGBD-based manipulations leveraging point cloud features. The proposed framework is simple and reasonable. I believe this work could have major impact in its field.

After rebuttal:

The rebuttal resolved most of my concerns in the review. I'd like to keep my recommendation at weak accept.

---

### Official Review · Reviewer_2cU3 · 2023-07-18

**Confidence:** 4
**Originality:** Good
**Technical Quality:** Very Good
**Clarity Of Presentation:** Very Good
**Impact:** 3

**Recommendation:**

Weak Accept: I recommend accepting the paper, but will not argue for my recommendation if the majority of other reviewers have a different opinion.

**Review:**

# Strength
* The paper is generally well-organized and clear to understand.
* The proposed method is easy to understand.
* Showed an elegant way of predicting action instead of discretizing the action space.
* The paper claims to introduce the first language-guided manipulation model that takes 3D pointcloud.
* Significantly compact/faster than the voxel-based approach of comparison.
* Detailed ablation tables are provided.

# Weakness
* Hard to understand the actual contribution of the model from the experiment.
  * For instance, Table 2 shows 10.5 percentage point difference with and without background/table removal. Without the removal, the proposed approach is outperformed by HiveFormer by 6.8 percentage point in the single task. No further information is provided for the multi-task setup. Based on the numbers in Table 2, seems like **the table removal contributed to the performance significantly**. For a fair comparison, **the empirical result of the approach without any removal should be included** or the equivalent removal should be provided to the 2D/3D models that are compared with.
  * In addition to that, it is hard to understand the types of variations in the task. For instance, if most of them are about background colors, the manual removal benefits more and relative contribution of the approach might be reduced.
* Hard to understand the benefit of using 3D pointcloud within the limited problem setup.
  * Without handling occlusion, it was hard to understand the benefit of using directly 3D pointcloud if it only applies to the setup with a table-top with fixed cameras. For instance, the approach that understands 3D geometry can be more robust with viewpoint variances than the 2D models as pointcloud embedding models are permutation invariant.
  * In comparison to the voxel-based approach, the proposed model shows similar constraints or limitation as those of the voxel-based
approach. It uses a down-sampled pointcloud with known a bounding-box and a table volume.

**Quality Of The Limitations Section:**

Limitations are addressed clearly

**Questions For Rebuttal:**

* How good the proposed method without background and table removal at the multi-task setup?
* Could you specify the variances of the tasks and provide statistics of category of variances?
* What would be the benefits of using the model that takes 3D pointclouds instead of RGBD images or voxels?
  * Would it be possible to show some experimental results that pointcloud-models can be useful than models with different input modalities?
* It is not clear what the caption of Figure 4 explains.

**Robotics Focus:**

Sufficient demonstration on hardware

**Summary Of Paper:**

The paper proposes a language-guided manipulation model that directly takes pointclouds and the authors show that a single model can handle multiple tasks with variations on the RLBench benchmark. It takes CLIP language embeddings and PointNext embeddings and uses a transformer to cross-attend for latent point feature to predict the next action with heatmap voting. The model was mainly compared with HiveFormer, a transformer-based approach with RGBD inputs, and PerAct, a voxel-based approach and showed better performance.

**Summary Of Recommendation:**

(Updated) I recommend a weak accept to the paper. A PointNet-based encoder is more efficient than the voxel-based approach and it may have a few advantages over the 2D or voxel-based approaches. After the rebuttal, the contribution from background/table removal became clear and showed the benefit of 3D model with camera perturbation along with a real-world experiment.

---

### Official Review · Reviewer_ABTj · 2023-07-23

**Confidence:** 3
**Originality:** Good
**Technical Quality:** Excellent
**Clarity Of Presentation:** Good
**Impact:** 3

**Recommendation:**

Weak Accept: I recommend accepting the paper, but will not argue for my recommendation if the majority of other reviewers have a different opinion.

**Review:**

Strengths:
- The paper performs a large series of experiments and ablations that can be useful for the reader, and any researcher that is designing point cloud-based robot manipulation.
- Results are strong, and although not superior to baselines in a statistically significant way, the reduction in computational complexity with respect to e.g. PerAct is still interesting.

Weaknesses:
- Claiming that the paper is the first to use a point cloud and language conditioned network may be excessive: while they are not entirely based on that, Hiveformer also uses point clouds in their network computation, and PerAct also uses voxelised point clouds, which are quite similar.
- The paper is not very clear in some parts, especially in the description of the network computation and its final output. If a reader is not familiar with the literature that uses point clouds, and the RLBench convention, it may be hard to follow at times.

I found it to be an interesting paper but the novelty of it may not be particularly significant, considering also that the results are not very different from Hiveformer. The point cloud is encoded through an existing architecture, PointNext, and the use of Transformers to manipulate tokens of different modalities has been explored several times.
I do believe however that the series of ablations are interesting and, although not all provide statistically significant results, they make the paper stronger.

**Quality Of The Limitations Section:**

Limitations are addressed clearly

**Questions For Rebuttal:**

I invite the authors to better position the novelty of their paper with respect to the existing baselines, that also use point clouds in a way or another. The paper can be strong even without the need for big novelty, but it is important to justify the strong claims made in the introduction.
While I realise it may be difficult to achieve during the short rebuttal phase, a few real world experiments may also make the paper stronger.

**Robotics Focus:**

Highly relevant to robotics but no hardware experiments

**Summary Of Paper:**

The paper introduces PolarNet, a point-cloud and language conditioned network for robot manipulation. It is based on a PointNext network to encode the point cloud, obtained by three cameras (right-left shoulder + wrist), and a Transformer to compute features taking into account both the encoded point cloud and the tokenized language command.
The authors compare their method with other state of the art methods on RLBench, a popular benchmark for robot learning in simulation, in different settings and with different tasks. Their experiments demonstrate in general comparable performance to PerAct and Hiveformer, two state of the art method which also use a combination of 2D and 3D inputs.
Additionally, the authors provide a series of ablations, analyzing many design choices involving the input space modality and point cloud processing, among the others.

**Summary Of Recommendation:**

I'm on the edge, and I will certainly discuss this with the other reviewers and the Area Chair after the rebuttal phase. While the results are convincing, the novelty is limited, and the paper feels quite similar in design to the baselines they mention. The thorough evaluation and ablations of their design choices are a plus, but there is also a lack of real world experiments.
**UPDATE**: after the rebuttal, I updated my score to weak accept.

---

### Official Review · Reviewer_mm9b · 2023-08-01

**Confidence:** 3
**Originality:** Good
**Technical Quality:** Good
**Clarity Of Presentation:** Very Good
**Impact:** 3

**Recommendation:**

Weak Accept: I recommend accepting the paper, but will not argue for my recommendation if the majority of other reviewers have a different opinion.

**Review:**

Strengths:
- The authors propose a new method that addresses the weaknesses of both 2D-image-based approaches (lack of precision) and 3D-based approaches (quantization, computation inefficiency)
- Ablations on the system design regarding the design choices are interesting

Weaknesses:
- No evaluation on real hardware. While simulation results can show whether an approach is promising, they don't always generalize well to the real world when it comes to perception or policy execution. It's hard to evaluate whether this approach shows promise on a real system.

**Quality Of The Limitations Section:**

Limitations are addressed clearly

**Questions For Rebuttal:**

While the existing simulation results show the efficacy of this approach, this paper would be greatly strengthened by compelling results on real hardware if that is possible in the revision period.

**Robotics Focus:**

Highly relevant to robotics but no hardware experiments

**Summary Of Paper:**

This paper proposes a policy called PolarNet that takes in 3D point clouds along with language instructions. A merged point cloud obtained from multi-view RGB-D cameras is encoded with a PointNext encoder, and a language instruction is encoded with the CLIP text encoder. These are given to a multi-layer transformer, and a PointNext decoder and an MLP together output the predicted action, which is an end effector pose along with gripper open state. The authors evaluate their approach on RLBench with 10 tasks under different settings, and compare their approach with baseline methods such as PerAct and Hiveformer. They show that their approach performs better and is more computationally efficient. They also run ablation studies of various components of their system.

**Summary Of Recommendation:**

The approach is interesing and shows good results in simulation but I am not convinced of the applicability to a real system if there is no hardware validation at all.

---

### Decision · Program_Chairs · 2023-08-30

**Decision:**

Accept (Poster)

**Comment:**

Reviewers felt that the paper presented a new method for a 3D point cloud policy with language instructions.  Reviewers appreciated the extensive experiments and ablations, including the recently added real-world experiments. Other parts of the paper were also greatly improved in the rebuttal.